# A Machine-Learning-Based Bibliometric Analysis of the Scientific Literature on Anal Cancer

**DOI:** 10.3390/cancers14071697

**Published:** 2022-03-27

**Authors:** Pierfrancesco Franco, Eva Segelov, Anders Johnsson, Rachel Riechelmann, Marianne G. Guren, Prajnan Das, Sheela Rao, Dirk Arnold, Karen-Lise Garm Spindler, Eric Deutsch, Marco Krengli, Vincenzo Tombolini, David Sebag-Montefiore, Francesca De Felice

**Affiliations:** 1Department of Translational Medicine (DIMET), University of Eastern Piedmont, 28100 Novara, Italy; marco.krengli@med.uniupo.it; 2Department of Radiation Oncology, “Maggiore della Carità” University Hospital, 28100 Novara, Italy; 3School of Clinical Sciences, Faculty of Medicine, Monash University, Clayton 3168, Australia; eva.segelov@monash.edu; 4Department of Oncology, Monash Health, Clayton 3168, Australia; 5Department of Hematology, Oncology and Radiation Physics, Skåne University Hospital, 22002 Lund, Sweden; anders.johnsson@med.lu.se; 6Department of Clinical Oncology, AC Camargo Cancer Center, São Paulo 01000-000, Brazil; rachelriechelmann@accamargo.org.br; 7Department of Oncology, Oslo University Hospital, 0316 Oslo, Norway; marianne.gronlie.guren@ous-hf.no; 8Institute of Clinical Medicine, University of Oslo, 0316 Oslo, Norway; 9Department of Radiation Oncology, The University of Texas MD Anderson Cancer Center, Houston, TX 77030, USA; prajdas@mdanderson.org; 10GI Unit, Royal Marsden Hospital, London SW3 6JJ, UK; sheela.rao@rmh.nhs.uk; 11Asklepios Tumorzentrum Hamburg, AK Altona, 22763 Hamburg, Germany; d.arnold@asklepios.com; 12Department of Oncology, Aarhus University Hospital, 8200 Aarhus, Denmark; k.g.spindler@rm.dk; 13Institute Gustave Roussy, 94805 Villejuif, France; eric.deutsch@gustaveroussy.fr; 14Radiation Oncology, Policlinico “Umberto I” and Department of Radiological, Oncological and Pathological Sciences, “Sapienza” University of Rome, 00161 Rome, Italy; vincenzo.tombolini@uniroma1.it (V.T.); francesca.defelice@uniroma1.it (F.D.F.); 15Leeds Institute of Medical Research, University of Leeds, Leeds LS2 9JT, UK; d.sebagmontefiore@leeds.ac.uk

**Keywords:** anal cancer, squamous-cell carcinoma, HPV, HIV, radiotherapy, oncology, bibliometrics, machine learning

## Abstract

**Simple Summary:**

Squamous-cell carcinoma of the anus, being a rare cancer, requires national and international collaborations, networking, organizational proficiency and leadership to overcome barriers towards the implementation of clinical trials to establish improved standards of care treatment strategies and the conduction of translational research projects to shed light into its biology and molecular characterization. The purpose of the present study is to obtain a global frame of the scientific literature related to anal cancer, through a bibliometric analysis of the published articles during the last 20 years (2000–2020), exploring trends and common patterns in research, tracking collaboration and networks to foresee future directions in basic and clinical research.

**Abstract:**

Squamous-cell carcinoma of the anus (ASCC) is a rare disease. Barriers have been encountered to conduct clinical and translational research in this setting. Despite this, ASCC has been a prime example of collaboration amongst researchers. We performed a bibliometric analysis of ASCC-related literature of the last 20 years, exploring common patterns in research, tracking collaboration and identifying gaps. The electronic Scopus database was searched using the keywords “anal cancer”, to include manuscripts published in English, between 2000 and 2020. Data analysis was performed using R-Studio 0.98.1091 software. A machine-learning bibliometric method was applied. The bibliometrix R package was used. A total of 2322 scientific documents was found. The average annual growth rate in publication was around 40% during 2000–2020. The five most productive countries were United States of America (USA), United Kingdom (UK), France, Italy and Australia. The USA and UK had the greatest link strength of international collaboration (22.6% and 19.0%). Two main clusters of keywords for published research were identified: (a) prevention and screening and (b) overall management. Emerging topics included imaging, biomarkers and patient-reported outcomes. Further efforts are required to increase collaboration and funding to sustain future research in the setting of ASCC.

## 1. Introduction

Squamous-cell carcinoma of the anus (ASCC) is considered a rare cancer, with an annual incidence of 0.5–2 new cases in 100,000 individuals [1]. Nevertheless, the incidence rate is constantly increasing, particularly in Europe, United States of America (USA) and Australia, mostly due to the human papilloma virus (HPV) epidemics in these regions [2]. Therefore, this clinical-pathological entity is regarded as a matter for global public health attention, with a focus on primary and secondary prevention, treatment and survivorship [3]. Given the rarity of the disease, barriers have been encountered when carrying out clinical trials to establish standard treatment strategies and to conduct translational research projects to shed light into its biology and molecular characterization [4,5]. However, despite these, ASCC has been a prime example of national and international collaborations, networking abilities, organizational proficiency and leadership. We performed a bibliometric analysis of ASCC-related literature of the last 20 years, exploring trends and common patterns in research, tracking collaboration and networks and foreseeing future directions in basic and clinical research. 

## 2. Materials and Methods

The electronic Scopus database was searched using the keywords “anal cancer”. Literature search was restricted to include manuscripts written in English and published between 1 January 2000 and 31 December 2020. The results of the electronic search were exported in a dataset and included citation information (authors, document title, year of publication, source title, volume, issue, pages, citation count, source and document type) and bibliographical information (affiliations, editors, keywords and funding details). Data analysis was performed using R-Studio 0.98.1091 software. A machine-learning bibliometric methodology was applied to evaluate the distribution of each factor. The bibliometrix R package was used [6]. “Summary ()” function was employed to summarize the main information regarding the bibliographic data, such as the annual percentage growth rate (in terms of relative percentage change between a later and an earlier time point), the most productive countries (based on first author’s affiliation), the country-specific production (based on authors appearance by country affiliations), the most relevant affiliations (based on disambiguated affiliation items, applying semantic similarity), the most-cited papers, the most-represented journals and the most frequent keywords. A factorial-analysis methodology was applied to identify joint keywords. The “conceptualStructure” function was used and multiple-correspondence analysis was applied to a *Document x Word* matrix [6]. The following options were used: method = “multidimensional scaling”, field = “Author’s keywords”, number of terms = “50”, number of clusters = “Auto”. Author’s keywords were plotted on a two-dimensional map and results were interpreted according to relative-point positions and their distribution along dimensions [6]. Scientific collaboration analysis was obtained using the “biblioNetwork ()” function that calculated a country collaboration network [NetMatrix <- biblioNetwork(M, analysis = “collaboration”, network = “countries”, sep = “;”)] [6]. International collaboration intensity of a specific country was based on the number of documents in which at least one coauthor worked in different country compared to the corresponding author [6]. “Biblioshiny ()” function was used to plot the global collaboration maps.

## 3. Results

### 3.1. Publication Numbers

A total of 2322 scientific documents were found, comprising articles (*n* = 1863), reviews (*n* = 281), letters (*n* = 83), conference papers (*n* = 58), editorials (*n* = 29) and short surveys (*n* = 8) (as per Scopus: short or mini-reviews of original research). There were fewer than 100 publications per year until 2010, then a significant increase in the subsequent decade, reaching a total of 223 manuscripts published in 2020. The average annual growth rate was around 40% over the period of analysis.

### 3.2. Mapping Scientific Collaboration

The five most productive countries were USA (*n* = 624), United Kingdom (UK) (*n* = 154), France (*n* = 124), Italy (*n* = 113) and Australia (*n* = 100). Country-specific production is displayed in Figure 1 (blue zone). Overall, publications originated from a total of 78 countries, with the top five being USA (*n* = 2428), France (*n* = 769), UK (*n* = 583), Australia (*n* = 578), and Italy (*n* = 482). 

In total, there were 47 countries represented by authors participating in the global anal-cancer literature (Figure 2). The USA and UK had the greatest link strength of international collaboration (22.6% and 19.0%, respectively). France, Australia and Italy had a high number of publications, but their international collaboration strength was relatively lower (13.6%, 16.9% and 8.0%, respectively). 

### 3.3. Keywords

As depicted in Figure 3, the factorial analysis identified two main clusters of keywords: (i) those mainly addressing screening and anal dysplasia; (ii) keywords linked to the overall management of anal cancer, from epidemiology and risk factors to treatment-related topics and survival.

After detailed analysis of the trend topics, we observed new items emerging in the last two years, including functional imaging [mostly, positron-emission tomography (PET)], biomarkers and patient-reported outcomes. To present a better visualization of the data records, a focus of a time span over ten years (2010–2020) is shown in Figure 4.

### 3.4. Journals

In the 20 years studied, 758 journals published articles on ASCC. Overall, 24% of journals (*n* = 182) have published more than two documents. The most represented journals were “Radiotherapy and Oncology”, “International Journal of Radiation Oncology Biology Physics” and “Diseases of the Colon and Rectum” with a total number of publications of 48, 47 and 44, respectively (see Figure 5). 

The three-field plot in Figure 6 reveals the relations between the most prolific countries and the relevant keywords and sources. The keyword parameter was used as a surrogate of research topic. The analysis of the top ten countries demonstrated that researchers around the globe had strong relations with the main topics of the literature [anal cancer, HPV, human immunodeficiency virus (HIV)] and had most frequently published within high-ranked international journals.

### 3.5. Affiliations and Citations

The institution with the highest number of contributions was the University of California, USA with a total of 129 documents, followed by the University of Melbourne, Melbourne, VIC, Australia (*n* = 65) and Mayo Clinic, Rochester, MN, USA (*n* = 52). It must be noted that despite the affiliation disambiguation, there are different locations indicating the same affiliation and the method may not precisely measure the similarities between affiliations. For instance, the contribution activity of the University of California includes different institutes’ resources, such as those located in San Francisco (UCSF), San Diego (UCSD), Los Angeles (UCLA), Irvine (UC Irvine) and Davis (UC Davis).

The distribution of institutes is reflected the authors’ productivity. The 10 most-cited papers (ranging 335 up to 856) are listed in Table 1 [7,8,9,10,11,12,13,14,15,16]. Among them, most documents referred to virus infection and absolute anal-cancer risk topics [7,8,9,10,13,15,16]. Treatment strategy was only studied in three papers [11,12,14]. Three papers were systematic reviews and meta-analyses [8,9,15]. The countries with the highest numbers of citations were France, the USA and the UK, with an average article citation of 31.66, 27.91 and 25.00, respectively.

## 4. Discussion

The annual scientific production published on ASCC is increasing, with an average annual growth rate around 40% and a steeper increase between 2010 and 2020 compared to the previous decade. Interestingly, amongst the more than 2000 publications published between 2000 and 2020, around 80% comprise original articles. A good proportion of them are published in high-ranked journals covering the field of clinical, radiation and gastrointestinal oncology. The geographical distribution covers all continents, which were generators of clinical and translational research. The predominant hubs for scientific production are the USA, Europe and Australia. Additionally, an active role is also played by Asian countries, despite the lower incidence of the disease in those areas, and by South America, particularly Brazil. Nations such as the USA and UK have both a high quantitative scientific throughput and an efficient collaborative international networking capacity, while others such as France, Italy and Australia can sustain high scientific productivity with a prevalent ‘standalone’ approach, focused more on national projects and collaboration. 

The geographical distribution of the scientific production is mirrored by the most productive academic institutions, which are located in the USA (University of California and Mayo Clinic) and Australia (University of Melbourne). One limitation of the current analysis is that collaboration was measured using the simple surrogate of coauthorship, which not necessarily reflects a well-functioning and active scientific network.

Nevertheless, collaboration and networking are crucial to promote clinical and translational research in a rare disease such as ASCC, where at present, appropriately funded national and international projects are scarce. However, in recent years, different initiatives have been settled and implemented to promote innovation in this setting. As an example, the International Rare Cancer Initiative (IRCI) for the relapsed/metastatic anal-cancer group completed the first multicentric randomized prospective controlled trial on advanced anal cancer, the InterAACT trial, which succeeded in patient recruitment across different collaborative groups in Europe [United Kingdom, Nordic Anal Cancer Group (NOAC), European Organization for Research and Treatment of Cancer (EORTC)], the US (NCI-endorsed initiative) and Australia (The Australasian Gastrointestinal Trials Group, AGITG), leading to a new standard of care for systemic therapy in metastatic ASCC [17]. The same cooperative group has been expanded and is now evaluating the role of anti- programmed cell death protein 1 (PD-1) inhibitor in inoperable, locally recurrent or metastatic ASCC not previously treated with systemic chemotherapy within a phase III, global and multicenter, double-blind randomized study (POD1UM-303/InterAACT2 trial) [18]. Other examples of cooperation in this clinical setting comprise the EORTC Gastrointestinal Tract Cancer Group (GITCG) Rectum, Anal Task Force and the National Cancer Institute’s National Clinical Trials Network (NCTN) Rectal–Anal Task Force. In the Nordic European countries, collaborations within NOAC have led to joint studies and the possibility to discuss advances in research and therapy [19].

In 2020 and 2021, clinicians and researchers organized two International Multidisciplinary Anal Cancer Conference (IMACC) Webinars. There was also a physical meeting held in Aarhus, Denmark, in 2021 [4,20]. These meetings presented an opportunity to discuss several potential topics and open issues for future collaborations, including the implementation of global prospective registries to record outcomes of specific patient subgroups, the harmonization of clinical-trial designs and outcomes to facilitate data pooling and analysis, and the need and rationale for developing biological and translational studies [4].

In our bibliometric evaluation, two main clusters of keywords were identified with the factorial analysis: the first being primary and secondary prevention, including screening and early detection; the second being overall therapeutic management. The last two decades have shed light into the causal effect of HPV infection as a prominent risk factor for precancerous lesions and invasive ASCC, together with information on the molecular pattern, biological characteristics and prognosis of HPV-related ASCC. The role of the HPV vaccine against anal HPV infection and anal intraepithelial neoplasia and its potential effect on the prevention of invasive cancer was also established [10]. Counterintuitively, the high number of publications related to the role of HPV in ASCC and the high quality of the scientific contributions published does not necessarily match the interest of the general community about this topic and the evidence of HPV-related cancer epidemics. A stronger effort should probably be made to raise awareness in the population about risk factors and sexual behaviors, to enhance prevention, screening and early detection and to advocate amongst policy makers for further dissemination of vaccination programs in young males and females [21]. With respect to the therapeutic management of ASCC, given that concurrent chemoradiation based on 5-fluorouracil/mitomycin-C (MMC) is the standard of care as demonstrated by first-generation trials [22,23,24], the last two decades reported the publication of three prospective randomized phase III trials studying potential new standard options. It was demonstrated that there is no benefit in substituting MMC to cisplatin, nor for induction chemotherapy, higher radiation-boost dose and maintenance chemotherapy [11,12,25]. Two of these trials are amongst the 10 most-cited publications in the study period [11,12]. Another highly cited document, reporting the results of the RTOG 0529 trial, explored the role of intensity-modulated radiotherapy (IMRT) to reduce the treatment-related acute toxicity profile [14]. Still, there have been relatively few large randomized trials to determine the optimal treatment regimens for ASCC. Currently, a large UK trial (PLATO) is investigating the optimal radiation doses for different clinical stages. 

It is interesting to note that a frequent focus has been on imaging, particularly magnetic resonance imaging (MRI) and fluorodeoxyglucose (FDG)-PET, highlighting the growing interest for these diagnostics tools in this setting. Pelvic MRI is now considered a mandatory examination, in agreement with the updated European Society for Medical Oncology (ESMO) guidelines, providing vital diagnostic, staging and prognostic information [5]. Moreover, FDG-PET is recommended, being potentially useful to confirm or exclude suspicious features on MRI, as well as to drive target volume selection and delineation for tailored IMRT approaches [5,26,27,28].

Biomarkers and patient-reported outcomes also emerged as recent trend topics in ASCC. The HPV/p16 status, tumor-infiltrating lymphocytes (TILs) rate, Epidermal Growth Factor Receptor (EGFR) overexpression, circulating HPV-DNA and blood biomarkers [Neutrophil-to-lymphocyte ratio (NLR), Systemic Index of Inflammation (SII), absolute number of leukocytes and neutrophils, hemoglobin level] were the most frequently studied biomarkers [29,30,31,32,33,34,35,36]. 

The Core Outcomes for clinical trials of chemoradiation for anal cancer (CORMAC) initiative in the UK and the work carried out within the EORTC Quality of Life (QoL) Group to develop the EORTC QLQ-ANL27 anal-cancer-specific questionnaire are valuable examples of the recent interest of the scientific community for patient survivorship and reported outcomes and QoL [37,38]. 

Of note, the terms related to late gastrointestinal function, sexual dysfunction and bone fracture do not appear as specific areas of previous research during the analyzed time period. Such data would, however, fuel the rationale for the needed clinical research aiming at decreasing sequelae, with treatment de-intensification and/or improved radiotherapy technologies [39,40].

Despite the relevant research achievements in ASCC during the last 2 decades, important gaps and unmet needs still exist, and there is a need to improve and increase international cooperation, under a shared vision and a global perspective. Some of the countries with a high incidence of ASCC (such as sub-Saharan Africa and South America) are underrepresented in the collaborative networks and very few studies on ASCC have been conducted enrolling patients in these regions. This observation hampers the generalizability of the clinical results observed in the randomized trials that set the standard of care for ASCC and call for external validation of these findings in underrepresented populations via a global effort. 

Few studies are available on equitable access to diagnosis and treatment for ASCC patients. Racial, gender and socioeconomic status disparities have been documented in this setting, and hence collaborative initiatives should be promoted to dissect causation, mitigate the impact and suggest potential operative solutions [41,42]. Similarly, for the subpopulation of HIV-positive ASCC patients, few clinical data are available highlighting the need for collaborative initiatives to fill the gap, such as the ACTION HIV lead by researchers in Brazil, to establish a global registry to collect clinical data and report on treatment outcomes in HIV ASCC patients [43].

## 5. Conclusions

We presented descriptive data from a bibliometric analysis on publications over the past 20 years on ASCC. Prevention and screening, diagnostics and biomarker landscapes, together with treatment strategies are the most explored topics in ASCC research. New topics are emerging, however. There is enthusiasm, precedence and growing output from international cooperation in the ASCC research field to promote advances and to identify and fulfill the unmet needs in this setting. It is finally important to highlight that there is an urgent need to increase collaboration and funding, to design proper clinical trials and implement timely translational-research projects to optimize treatment and improve clinical outcomes in patients with ASCC.

## Figures and Tables

**Figure 1 cancers-14-01697-f001:**
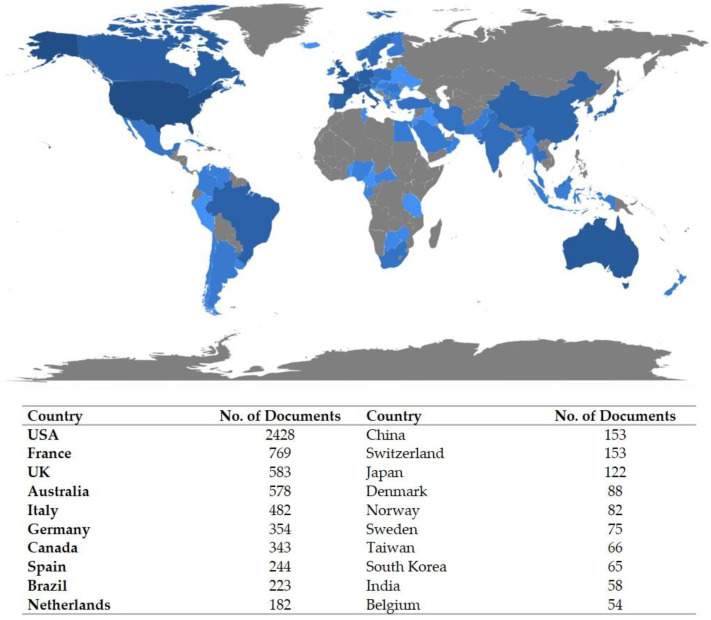
Country-specific production. The color intensity is proportional to the number of documents. Higher blue intensity refers to greater number of documents (dark blue = high productivity; grey = no documents).

**Figure 2 cancers-14-01697-f002:**
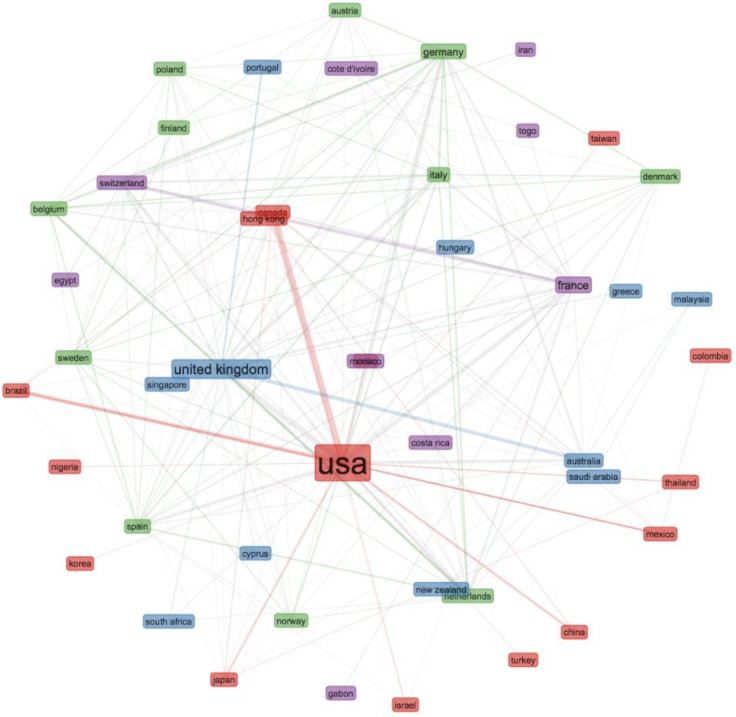
Collaboration network. The size of the rectangle is proportional to the number of publications; the thickness of line represents the strength of collaborations; the colors represent the collaboration clusters.

**Figure 3 cancers-14-01697-f003:**
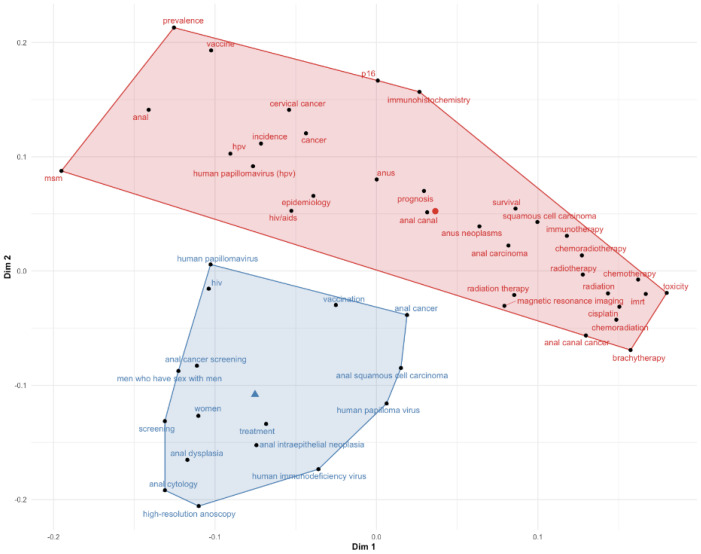
Factorial analysis of analyzed keywords. The distance between keywords indicates shared substance. The colors distinguish different clusters.

**Figure 4 cancers-14-01697-f004:**
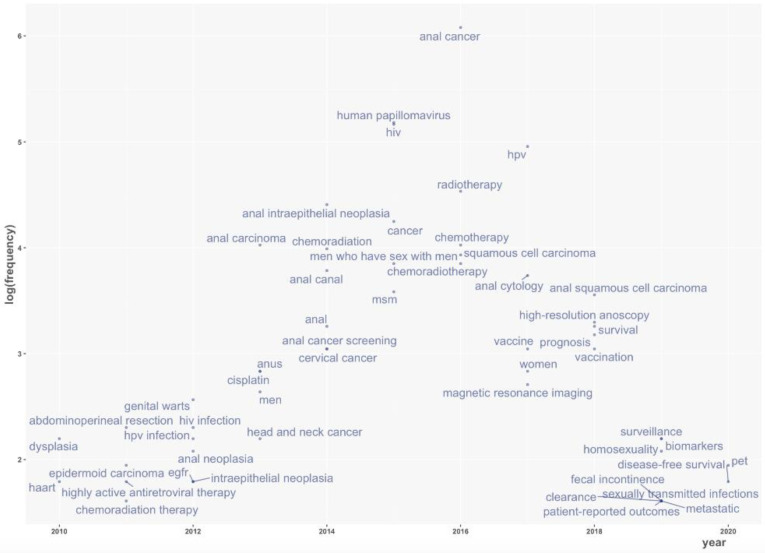
Trend topics in the last 10 years. The graph represents the use of the most frequent keywords over time; the respective frequency (for the specific year) is displayed on a vertical axis in a logarithmic scale. Each keyword is marked on the year with its highest prevalence in the literature.

**Figure 5 cancers-14-01697-f005:**
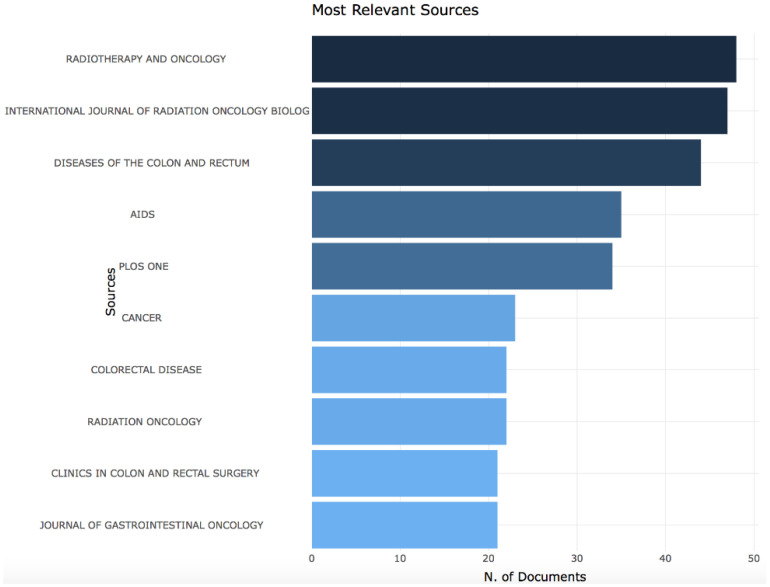
Most relevant journals.

**Figure 6 cancers-14-01697-f006:**
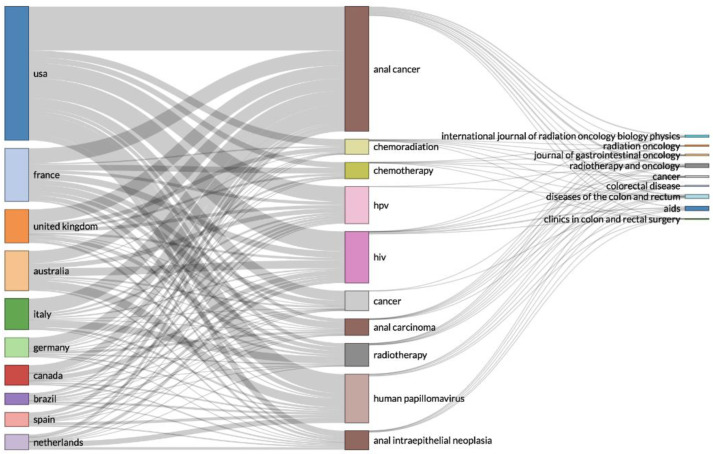
Three-field plot of classification by country, keyword and source from all records.

**Table 1 cancers-14-01697-t001:** Most-cited documents.

References	Title	Total Citations *	Total Citations per Year **
Enels EA, 2011, JAMA [7]	Spectrum of cancer risk among US solid organ transplant recipients	856	78
De Vuyst H, 2009, Int J Cancer [8]	Prevalence and type distribution of human papillomavirus in carcinoma and intraepithelial neoplasia of the vulva, vagina and anus: a meta-analysis	684	53
Machalek DA, 2012, Lancet Oncol [9]	Anal human papillomavirus infection and associated neoplastic lesions in men who have sex with men: a systematic review and meta-analysis	621	62
Palefsky JM, 2011, New Engl J Med [10]	HPV vaccine against anal HPV infection and anal intraepithelial neoplasia	615	56
Ajani JA, 2008, JAMA [11]	Fluorouracil, mitomycin, and radiotherapy vs. fluorouracil, cisplatin, and radiotherapy for carcinoma of the anal canal: a randomized controlled trial	594	42
James RD, 2013, Lancet Oncol [12]	Mitomycin or cisplatin chemoradiation with or without maintenance chemotherapy for treatment of squamous-cell carcinoma of the anus (ACT II): a randomised, phase 3, open-label, 2 × 2 factorial trial	395	44
Silverberg MJ, 2012, Clin Infect Dis [13]	Risk of anal cancer in HIV-infected and HIV-uninfected individuals in North America	369	37
Kachnic LA, 2013, Int J Radiat Oncol Biol Phys [14]	RTOG 0529: a phase 2 evaluation of dose-painted intensity modulated radiation therapy in combination with 5-fluorouracil and mitomycin-C for the reduction of acute morbidity in carcinoma of the anal canal	351	39
Shiels MS, 2009, J Acquired Immune Defic Syndr [15]	A meta-analysis of the incidence of non-AIDS cancers in HIV-infected individuals	350	27
Giuliano AR, 2011, Lancet [16]	Incidence and clearance of genital human papillomavirus infection in men (HIM): a cohort study	335	31

* Number of times each document has been cited; ** yearly average number of times each document has been cited.

## Data Availability

The data presented in this study are available on request from the corresponding author.

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
