# Peer review of "A Machine-Learning-Based Bibliometric Analysis of the Scientific Literature on Anal Cancer"

_cancers, 2022, doi:10.3390/cancers14071697_

Round 1

Reviewer 1 Report

The work concerns the bibliometric analysis of publications related to a rare cancer which is anal cancer. This analysis should be published in a journal dedicated to such analyzes.

Reviewer 2 Report

No comments

This manuscript is a resubmission of an earlier submission. The following is a list of the peer review reports and author responses from that submission.

Round 1

Reviewer 1 Report

This is a bibliometric work and the presented data does not bring any particularly interesting information. The authors focused on citations, the country of origin and the type of the journal. There are no references to the number of patients or the specific research subject. There are no references to epidemiology and disease statistics in these countries. The analysis adds little to the knowledge of this subject.

Author Response

We would like to thank the reviewers for their (mostly) positive comments regarding our manuscript and for the helpful suggestions that we implemented within the text in order to optimize the revised draft, that we are hereby enclosing.

 This is a bibliometric work and the presented data does not bring any particularly interesting information. The authors focused on citations, the country of origin and the type of the journal. There are no references to the number of patients or the specific research subject. There are no references to epidemiology and disease statistics in these countries. The analysis adds little to the knowledge of this subject.

The idea behind the present study, was provide a snapshot of the most recent (2000-2021) scientific literature on anal cancer to explore trends and common patterns in research, tracking collaboration and networks to foresee future directions in basic and clinical research. We hence focused on the scientific throughput as target of the analysis, knowing that it may be a mirror of clinical needs, research resource allocation and research trends. The reason why epidemiology and disease statistics were not referenced is because they were NOT a topic of the present analysis, which was based on bibliometric parameters, to represent what the scientific community was able to produce in the 2 decades analyzed.

Reviewer 2 Report

A nice report, and an overall pleasant read. The figures look highly informative and well presented. My objection is with the discussion that looks and feels unnecessarily long for such a document. Please shorten.

Author Response

We would like to thank the reviewers for their (mostly) positive comments regarding our manuscript and for the helpful suggestions that we implemented within the text in order to optimize the revised draft, that we are hereby enclosing.

A nice report, and an overall pleasant read. The figures look highly informative and well presented. My objection is with the discussion that looks and feels unnecessarily long for such a document. Please shorten.

Thank you for the suggestions. We modified the discussion accordingly and shorthened the text.

Reviewer 3 Report

Very well conceived, written and discussed article. Both topic and methods are very timely in terms of analytical process. Clearly AI and ML are expanding their scope in all aspects of information technology. The data analyzed was more "demographic" than actual oncologic data points. An interesting and more valuable application of AI and ML methods may be applied to National Cancer Data Base information, possibly compared across different countries and health care systems.Nonetheless, a worthwhile effort that should prompt additional applications as suggested above.

Author Response

We would like to thank the reviewers for their (mostly) positive comments regarding our manuscript and for the helpful suggestions that we implemented within the text in order to optimize the revised draft, that we are hereby enclosing.

Very well conceived, written and discussed article. Both topic and methods are very timely in terms of analytical process. Clearly AI and ML are expanding their scope in all aspects of information technology. The data analyzed was more "demographic" than actual oncologic data points. An interesting and more valuable application of AI and ML methods may be applied to National Cancer Data Base information, possibly compared across different countries and health care systems.Nonetheless, a worthwhile effort that should prompt additional applications as suggested above.

Thank you for you comments. In the present study we focused on bibliometric data to explore trends and common patterns in research, tracking collaboration and networks to foresee future directions in basic and clinical research.